# Impact of Microplastics on the Ocular Surface

**DOI:** 10.3390/ijms24043928

**Published:** 2023-02-15

**Authors:** Duoduo Wu, Blanche X. H. Lim, Ivan Seah, Shay Xie, Julia E. Jaeger, Robert K. Symons, Amy L. Heffernan, Emily E. M. Curren, Sandric C. Y. Leong, Andri K. Riau, Dawn K. A. Lim, Fiona Stapleton, Mohammad Javed Ali, Swati Singh, Louis Tong, Jodhbir S. Mehta, Xinyi Su, Chris H. L. Lim

**Affiliations:** 1Department of Ophthalmology, National University Health System, Singapore 119228, Singapore; 2Yong Loo Lin School of Medicine, National University of Singapore, Singapore 119077, Singapore; 3Eurofins Environment Testing Australia & New Zealand, Dandenong, VIC 3175, Australia; 4Australian Water Association, St Leonards, NSW 2065, Australia; 5St. John’s Island National Marine Laboratory, Tropical Marine Science Institute, National University of Singapore, Singapore 119077, Singapore; 6Singapore Eye Research Institute, Singapore 169856, Singapore; 7Ophthalmology and Visual Sciences Academic Clinical Programme, Duke-NUS Medical School, Singapore 169857, Singapore; 8School of Optometry and Vision Science, University of New South Wales, Sydney, NSW 2052, Australia; 9LV Prasad Eye Institute, Hyderabad 500034, India; 10Singapore National Eye Centre, Singapore 168751, Singapore

**Keywords:** microplastics, ocular surface, dry eye disease, polymers, dysbiosis, oxidative stress, inflammation

## Abstract

Plastics are synthetic materials made from organic polymers that are ubiquitous in daily living and are especially important in the healthcare setting. However, recent advances have revealed the pervasive nature of microplastics, which are formed by degradation of existing plastic products. Although the impact on human health has yet to be fully characterised, there is increasing evidence that microplastics can trigger inflammatory damage, microbial dysbiosis, and oxidative stress in humans. Although there are limited studies investigating their effect on the ocular surface, studies of microplastics on other organs provide some insights. The prevalence of plastic waste has also triggered public outcry, culminating in the development of legislation aimed at reducing microplastics in commercial products. We present a review outlining the possible sources of microplastics leading to ocular exposure, and analyse the possible mechanisms of ocular surface damage. Finally, we examine the utility and consequences of current legislation surrounding microplastic regulation.

## 1. Introduction

Plastics are synthetic or semi-synthetic polymers. Their use is ubiquitous in daily living and particularly in the healthcare setting ranging from packaging to equipment manufacturing. Despite its utility, there have been concerns over the adverse health effects of plastics.

All plastics undergo degradation into smaller particles, termed “microplastics” and “nanoplastics”. The term “microplastic” was first coined in 2004 to describe microscopic plastic particles in marine sediments [1]. These exist as either primary or secondary microplastics. Primary microplastics are manufactured microplastics of small size, such as microbeads and resin pellets. Secondary microplastics originate from the breakdown of larger plastic particles such as plastic bottles or bags due to the action of physical, chemical, and biological degradation. However, there is increasing evidence that microplastic particles are omnipresent and have been observed in the air, water, food, and, more recently, in humans [2]. The definition of microplastics and nanoplastics constantly evolves vis-à-vis our growing understanding of their presence and impact. The European Chemicals Agency (ECHA) defines microplastics as “solid-polymer containing particles, to which additives or other substances may have been added, and where ≥1% weight by weight of particles have (i) all dimensions 1 nm ≤ x ≤ 5 mm, or (ii), for fibres, a length of 3 nm ≤ x ≤ 15 mm and length to diameter ratio of >3”, while Gigault et al. define nanoplastics as “particles unintentionally produced (i.e., from the degradation and the manufacturing of the plastic objects) and presenting a colloidal behaviour, within the size range varies from 1 to 1000 nm” [3].

Microplastics have a high surface-area-to-volume ratio and bioaccessibility due to their small size, allowing them to exert effects on human health at a cellular level [4]. Exposure to microparticles triggers microbial dysbiosis, oxidative stress, and chronic inflammation in the human body. These particles have also been implicated in malignancies and may also affect foetal development [5,6,7]. Microplastics can also act as chemical and pathogen vectors that may exert both toxic and hormonal influences on the human body [8]. Recently published experiments in murine models have demonstrated that microplastics can stimulate ocular surface inflammation and damage, induce apoptosis, and reduce corneal and conjunctival epithelial cell viability [9,10].

This review outlines possible sources of ocular surface exposure to microplastics, the impact on the ocular surface, and proposed mechanisms of damage. Current quantification methods for microplastics and nanoplastics, along with their limitations, will also be discussed. Lastly, this review provides a summary of existing regulations governing the manufacturing and monitoring of microplastics.

## 2. Sources and Routes of Exposure to Microplastics

Possible environmental sources include exposure to microplastic-contaminated fluids or air (Figure 1). The most common microplastics found in the environment include polypropylene, polyethylene, polystyrene, and polyethylene terephthalate [4]. Estimates of the half-life of these particles vary according to their polymer components, environmental factors, and thickness [11,12]. For instance, low density polyethylene has an average half-life of 4.6 years when buried, and 3.4 years in the marine environment. The half-life is further shortened in the presence of environmental factors such as heat and ultraviolet irradiation [12]. Exhaust gas from motor vehicles contains air-borne microplastics [13]. Other sources include incinerators, landfills, industrial emissions, agricultural fertiliser as well as synthetic textiles [14]. The daily washing of synthetic textiles in a household releases at least half a million microfibres from each kilogram of clothing [15].

Shedding of microplastics from household items has also been reported. There is evidence of microplastics released from plastic food containers, disposable cups, and plastic tea filter bags [16,17,18]. Contamination of raw food via contact with plastic chopping boards has also been demonstrated [19]. Multiple factors such as heat and physical stress can influence the volume of microplastics shed from plastic materials. When a polypropylene infant milk bottle was heated from 25 to 90 °C, the number of microplastics released increased from 0.6 to 55 million particles per litre [20].

Microplastics have also been retrieved from the surgical environment and are thought to arise from the abundant use of plastics in the healthcare setting [21]. Given widespread reports of microplastics arising from common plastic packaging and everyday items such as bottled water, it is possible that microplastics may be present in eye drops [22]. This is of concern as patients with chronic diseases such as dry eye disease and glaucoma, where frequent and prolonged eye drop instillation is required, may unwittingly expose the ocular surface to microplastics within topical ophthalmic formulations. This is an important but underappreciated exposure.

## 3. Impact of Microplastics on the Ocular Surface

In vitro studies showed both human cornea and conjunctival epithelial cell lines could take up polystyrene microplastic particles with microplastics accumulating around the cell nuclei. These particles were cytotoxic, with decreased cell viability and proliferation markers identified [9]. To explore the impact of exposure of the ocular surface to microplastics in murine models, test mice received 2.5 μL of a topical suspension containing 1 mg/mL of either 50 nm or 2 μm polystyrene microplastics three times a day without anaesthesia, for two to four weeks [9]. The control group was similarly treated with normal saline and another ‘normal’ group did not receive any interventions.

Ocular surface fluorescein staining was evaluated weekly and increased staining was observed in the test group but not in the control or normal group. Interestingly, sporadic punctate staining was seen in the group of mice receiving administration of normal saline. There was no mention of how the normal saline was stored (presumably in a plastic bottle); neither was testing of the normal saline solution for microplastics described beforehand. Tear film secretion was investigated weekly with a phenol red thread test. A reduction in tear secretion was identified and tear secretion reduced over time in the two treatment groups.

Stereo-fluorescence microscopy further demonstrated accumulation of microplastic particles in the lower conjunctival sac that increased over time. Analysis of ex vivo tissues at the end of the study showed reduced size and density of goblet cells of the lower lid compared to the control group. Proliferation-related markers (Ki-67, p63, and K14) were also downregulated in the treatment groups compared to controls.

Compared to the normal and control groups, there was an irregular arrangement of lacrimal gland acini in both treatment groups. Inflammatory cells between acini and upregulation of inflammatory factors and cytokines (IL-1α, IL1-β, and IL-6) in a time-dependent fashion were also reported. There were higher rates of apoptosis identified in mice receiving the suspension containing 50 nm compared with 2 μm microplastic particles [9].

Exposure of the murine ocular surface to particulate matter 2.5 (PM2.5) environmental pollutants, which can contain microplastics, causes reduced tear volume, tear film break-up time, and destruction of corneal epithelial microvilli and corneal desmosomes [10]. Increased levels of TNF-α and NF-κB p65 (Ser-536 phosphorylation) on the ocular surface suggested ocular surface disorders similar to human dry eye disease. A prospective multicentre cohort study of 387 dry eye disease patients in China noted worse Ocular Surface Disease Index (OSDI) scores, meibomian gland dysfunction, and increased levels of IL-8 and IL-6 in regions with higher PM2.5 levels [23]. Similar observational studies have also demonstrated exacerbation of ocular surface instability and dry eye disease with exposure to environmental pollutants [24,25].

Although limited studies have been performed on the ocular surface, much work has been performed in other organs which may predict the impact of microplastics on the ocular surface. Tissue and cell damage may be caused by: (1) inflammation and oxidative damage, (2) microbial dysbiosis, and (3) toxicological effects from additives and sequestrated compounds.

## 4. Proposed Mechanisms of Tissue and Cell Damage

### 4.1. Inflammation and Oxidative Damage

While the role of inflammation in patients with dry eye disease is well-established, our understanding of how it fits into pathological mechanisms remains controversial. For instance, key differences in how inflammation arises and perpetuates in patients with dry eye disease remain unknown [26]. Regardless, inflammation is recognised as a key contributory and exacerbating factor in the pathogenesis of dry eye disease.

Oxidative stress can cause inflammation and dry eye disease [27]. Dry eye individuals have higher levels of late lipid peroxidation markers including 4-hydroxynonenal (4-HNE) and malondialdehyde (MDA), which are indicative of oxidative stress [28]. Importantly, MDA levels correlated with increased disease severity (worse tear film break-up time, Schirmer’s test, conjunctival goblet cell density and symptoms). In particular, the production of reactive oxygen species (ROS) was associated with inflammatory cell infiltration over the ocular surface [29].

Similar oxidative damage can be induced by microplastic exposure. Microplastic uptake in the intestinal system is governed via microvilli endocytosis and cilia movement, which transfers these particles into digestive tubules [30]. Knowledge surrounding the mechanisms of cellular uptake of microplastics, and their eventual outcome remains limited [31]. Intestinal exposure to microplastics in invertebrates (*Mytilus* spp., *Caenorhabditis elegans*, *Artemia parthenogenetica*) increased intestinal expression of glutathione S-transferase 4 and lipid peroxidation, and reduced catalase and glutathione reductase, suggesting that oxidative damage is a key mechanism in microplastic-induced epithelial damage [32,33,34].

Oral ingestion of microplastics in aquatic vertebrates (*Danio rerio*, *Poecilia reticulata*, *Girella laevifrons*, *Larimichthys crocea*) increased intestinal levels of TNF-α, IFN-γ, IL-1α, IL1-β, and IL-6 [35,36]. Manifestations include goblet cell enlargement, leukocyte infiltration, reduced digestive enzyme activity, and the loss of intestinal villi and crypt cells [37,38]. In mice, gut exposure to microplastics significantly increased expression of toll-like receptor 4 (TLR4), activator protein-1 (AP-1), and interferon regulatory factor 5 in the colon and duodenum, all of which are associated with inflammation [39]. Moreover, reduced mucus secretion, impaired intestinal permeability, and histological inflammation in the duodenum and colon were observed [36,40,41].

Cytotoxic responses after exposure to microplastics have also been reported in the respiratory tract. Flock worker’s lung, for example, is an occupational disease attributed to airway exposure to polyethylene, polypropylene, and rayon flock fibres, resulting in the development of restrictive lung disease [42]. Microplastics induce cytotoxic and inflammatory effects in human lung epithelial cells (BEAS-2B), in vitro, through the formation of reactive oxygen species (ROS) [43]. Exposure of lung epithelial cells to increasing concentrations of microplastics causes increased proinflammatory cytokine levels, epithelial cell apoptosis, and increased severity of epithelial damage [44]. Moreover, 80 nm microplastics can also cause mitochondrial damage by penetrating human hepatic (L02) and lung (BEAS-2B) and causing overproduction of mitochondrial ROS while suppressing mitochondrial respiration [45].

Apart from direct inflammatory damage, there is evidence that nanoplastics result in cellular damage by fundamentally altering protein structures. A study using molecular dynamic simulations showed that 5 nm polyethylene nanoplastics increased the presence of protein α-helices, while nylon nanoplastics induced the unfolding of helical structures and promoted the formation of β-sheet structures [46]. This suggests that nanoplastics may interact with secondary protein structures and is postulated to be a potential cause of amyloidosis, a key process implicated in diseases such as Parkinson’s disease and Alzheimer’s disease [47].

These findings suggest that despite plastics being generally innocuous to the general population, exposure of the cellular environment to microplastics and nanoplastics can invoke an inflammatory and cytotoxic response in tissues and induce cellular damage. The inflammatory reactions are driven by both innate and adaptive immune responses, as evidenced by the expression of TLR4 and adaptive-response cytokines TNF-α, IFN-γ, IL-1α, IL1-β, and IL-6 [37,38,39]. On the ocular surface, similar pro-inflammatory states were observed when the ocular surface of mice were exposed regularly to microplastics; IL-1α, IL1-β, and IL-6 were upregulated in the conjunctiva and lacrimal glands [9]. More studies are required to further characterise the impact of microplastics on the human ocular surface.

### 4.2. Microbial Dysbiosis

Ocular surface diseases have been associated with alterations in the ocular surface microbiome. As an example, the ocular surface of patients with aqueous tear-deficient dry eye possesses an increased abundance of *Brevibacterium* and a reduced amount of *Pseudomonas*, while patients with dry eye disease associated with meibomian gland dysfunction have a higher abundance of *Firmicutes* and *Proteobacteria*, and reduced levels of *Actinobacteria* [48,49]. Additionally, the conjunctiva microbiome differs between participants from three cities, showing that environmental factors such as climate and pollution could play a role [50]. The ocular surface microbiome is likely to maintain homeostasis and modulate ocular surface immune function and may therefore play an important role in the pathogenesis and development of ocular surface diseases [51].

In vivo metagenomic studies in various species of vertebrates have shown that microplastics can cause microbial dysbiosis. Intestinal exposure to polystyrene particles reduced bacterial biodiversity in Zebrafish. The affected population of bacteria varies across different studies—while a study showed a decreased abundance of *Proteobacteria* with increased levels of *Firmicutes*, another study showed decreased *Actinobacteria* population, but increased *Proteobacteria* levels [36,52,53]. Gut exposure to polyethylene in murine models induces a significant increase in *Staphylococcus* population and a drop in *Parabacteroides* abundance [39].

An in vitro study using Simgi^®^, a computer-controlled simulator of the human gastrointestinal tract, reported morphological changes to microplastic particles after gastrointestinal digestion and colonic fermentation [54]. Biodegradation of microplastic particles through the gastrointestinal system led to the deposition of organic matter and colonic microbiota on the surfaces of microplastic particles. In particular, this study also showed that populations of *Bacteroides*, *Parabacteroides*, and *Alistipes* dropped while populations of *Escherichia*, *Shigella*, and *Bilophila* rose in the colon following exposure to microplastics [54].

Evidence of alteration in bacterial abundance from these in vitro studies suggests that microplastics may alter the local microbial environment such that specific populations are adversely affected. Crucially, the severity of inflammatory bowel disease (ulcerative colitis) has been associated with lower levels of *Parabacteroides* in faeces, implying that microplastics may stimulate microbial dysbiosis and exacerbate intestinal diseases [55,56].

A similar dysbiotic impact of microplastics has also been observed in non-gastrointestinal systems. A study analysing the microplastic and microbiome composition in human placenta and newborn meconium samples retrieved 16 different types of microplastics from all samples [57]. Results from this investigation showed that microplastic levels were inversely proportional to the abundance of *Parabacteroides* in the meconium (ethylene-vinyl acetate) and *Bacteroides* in the placenta (polyethylene), suggesting that specific microplastics may impact upon the viability of various microbiota.

Contact lens wear is associated with ocular surface microbial dysbiosis with a shift towards periorbital skin biota [58]. Additionally, the type of contact lens may impact ocular surface biota, where orthokeratology lens wearers have significantly less *Bacillus*, *Tatumella*, and *Lactobacillus* species while soft contact lens users had lower abundance of *Delftia* and more *Elizabethkingia* than non-contact lens wearers [59]. These differences in ocular microbiota have been suggested to be related to mechanical pressure and hypoxia [60]. No studies have examined the impact of microplastics on the ocular microbiome. It remains to be investigated if a similar relationship between microplastic-induced microbial dysbiosis and severity of the ocular surface diseases exists.

### 4.3. Toxicological Effects of Additives and Sequestrated Compounds

Microplastics and their degradation products may harbour toxic chemicals arising from either additives during the manufacturing process, or chemicals absorbed by plastics from the environment [61]. Plastics can hyper-concentrate chemical additives and compounds absorbed from their surroundings. A study identified 1411 unique chemical compounds extracted from common daily plastic consumer products, including bottles, slippers, floor covering, and trays [62]. Extracts revealed varying levels of estrogenicity, anti-androgenicity, oxidative stress responses, and cytotoxicity.

Additives are chemicals incorporated into plastics during production to augment their properties, such as colour, transparency, and durability. There are numerous additives of concern that have the potential to damage human tissues. Phthalates are esters of phthalic acid (1,2-benzene dicarboxylic acid) used to produce polyvinyl chloride (PVC). Epidemiological studies have identified phthalates as key culprits of suppressed reproductive hormones, altered thyroid function, and the development of obesity and metabolic syndrome [53,54,55,56,57,58,59,60,61,62,63,64,65,66,67,68]. Exposure of human corneal endothelial cells (B4G12 cell line) to phthalates increased the production of IL-1β, IL-6, and IL-8, manifesting as decreased cell proliferation and subsequent cell toxicity [69]. Human lens epithelial cells experience a dose-dependent loss of viability when exposed to phthalate, even at low concentrations [70]. Another common additive is Bisphenol A (BPA), which is formed via the condensation of phenol and acetone, and is used in the production of polycarbonates. BPA induces oxidative stress in tissues, causing mitochondrial damage and cell apoptosis [71]. It is also associated with an increased risk of cancer, cardiovascular disease, and reproductive disorders [72,73,74,75].

Heavy metal additives incorporated into plastics to imbue specific properties can also potentiate diseases. Cadmium, mercury, and arsenic can induce carcinogenesis, while copper and cobalt can induce the formation of ROS [76,77,78,79,80]. Heavy metals have been also shown to be associated with dry eye disease. A large population study in Korea demonstrated an association between the detection of serum mercury and development of dry eye disease among females, while a cross-sectional study of welders in Taiwan reported associations between high levels of urinary cadmium and toenail lead concentrations, and dry eye disease [81,82].

Plastics sequester other toxic chemicals and heavy metals from their surroundings [83,84,85]. Organic pollutants are sequestrated in hydrophobic plastics due to their low water solubility and high fat solubility [86,87]. A study comparing the effectiveness of five multipurpose contact lens solutions against *Pseudomonas aeruginosa*, *Staphylococcus aureus*, and *Fusarium solani* showed that preincubation of solutions with contact lenses led to a decrease in effectiveness against the bacterial strains [84]. This was postulated to have occurred either due to inactivation or absorption into the lens [84]. Storage of chlorhexidine gluconate (a preservative used in rigid contact lens solutions) in polyethylene and polypropylene containers also resulted in up to 12% loss of chlorhexidine concentration over time due to adsorption [85]. Microplastics possess a large surface area which further augments their ability to absorb toxic chemicals.

Unfortunately, our understanding of the interactions between plastics and various chemicals remains limited, and the vast majority of chemicals isolated from plastic compounds remain unidentified and unstudied [88]. Current studies on additives found in plastics represent a minority of characterised compounds, with more studies urgently required to fully characterise the health impact of these chemicals on human health.

## 5. Recommendations for Testing of Microplastics and Limitations of Current Methods

Despite ongoing progress, this field remains relatively new and challenges in accurate identification and quantification of microplastics exist. This section describes common protocols for sample processing and analysis, with an emphasis on predictability, reproducibility, and accuracy of methods.

### 5.1. Laboratory Protocols

Prevention of environmental contamination during sample preparation and analysis is important. The proposed methodology to reduce plastic contamination includes utilising non-plastic apparatus such as glass or steel devices, deep cleaning of work stations prior to experiments with plastic-free ethanol and Milli-Q water, wearing natural fibre clothing, minimising movement of personnel in and out of the laboratory, and keeping a database of plastics that may come into contact with the samples [89,90]. Experiments should also be conducted in a laminar flow box if possible, and samples covered with aluminium foil or glass to avoid contamination [91]. The Baselines and Standards for Microplastics Analyses in European Waters (BASEMAN) projects recommend running a minimum of three procedural blanks (distilled water) treated with the same procedure and chemicals to assess for baseline microplastic contamination [92].

### 5.2. Pretreatment

Pretreatment aims to remove contaminants which may confound subsequent analyses. To separate inorganic contaminants from microplastics, samples are first filtered and subsequently added to solutions (such as sodium chloride, sodium polytungstate, sodium iodide, calcium chloride, and zinc chloride) as vehicles to separate microplastics from the original sample via density differences [93,94,95]. Additional methods such as centrifugation and air bubbling have also been utilised to reduce the processing time [95].

Acid-base solutions such as hydrogen peroxide and potassium hydroxide are used to remove organic materials [96]. A major limitation of these reactants is the concurrent degradation of microplastic particles in the original sample and alteration of their characteristics. Although the use of sodium hydroxide removed two-thirds of organic matter from sludge and soil samples, it led to significant degradation of polyethylene terephthalate and polycarbonate particles [97]. Hydrogen peroxide decreases the rates of recovery of microplastics and alters the colour of microplastic fragments [98]. Incubation with potassium hydroxide at 40 °C eliminates organic materials while being inert towards plastic polymers [98]. Promisingly, enzyme digestion techniques remove organic materials without damaging microplastic particles [99,100]. However, this method is costly and time-consuming as it takes days, depending on the extent of the contamination.

### 5.3. Microplastic Analysis Methods

Following pretreatment, further analysis to quantify and characterise microplastics is performed, with common methods summarised in Table 1. These can be broadly divided into non-destructive and destructive analytic approaches.

#### 5.3.1. Non-Destructive Methods

##### Light Microscopy

Visual identification of microplastics can be conducted with visual, light, or digital microscopes. This method is preliminary, operator dependent, subjective, and associated with high rates of misidentification and the inability to detect very small sizes (the smallest microplastic size reliably identified is 500 μm due to high rates of misidentification) [101,102]. Hence, it is often performed in conjunction with more sophisticated analyses.

##### Stereomicroscopy

The stereomicroscope allows three-dimensional visualisation of microplastic particles by allowing observation of the sample from two different angles. Although this is limited by a lower magnification compared to conventional light microscopy, it provides a better visual characterisation of the surface structure and morphology of microplastic particles. Unfortunately, the usage of this technique is limited by the quality of samples—samples with impurities which cannot be chemically digested and samples with thick, dense sediment often makes visualisation difficult. Previous studies have shown a 20–70% identification rate of transparent particles using stereomicroscopy when validated against other techniques [103,104].

##### Fluorescence Microscopy

Fluorescence microscopy, such as staining the sample with Nile Red reagent, can highlight microplastics and assess the count and nature of microplastics [105]. This method can be combined with conventional light microscopy and stereomicroscopy to provide better visualisation and identification of microplastics in a sample. A limitation of Nile Red reagent staining is co-staining of natural organic material—hence, adequate and thorough pretreatment is required during sample preparation [106].

##### Transmission Electron Microscopy

Transmission electron microscopy (TEM) is one of the most commonly utilised techniques in characterising nanoparticles as it provides chemical information and imagery at atomic resolutions [107]. However, TEM is ineffective at visualising nanoplastics due to their amorphous structure and electron-lucent nature [108].

##### Scanning Electron Microscopy

Scanning electron microscopy (SEM) with energy dispersive X-ray spectroscopy (EDS) can analyse the chemical composition of particles [109]. SEM-EDS has been used to study and identify nanoplastics, however it is unable to distinguish synthetic nanoplastics from natural non-plastic nanoparticles, especially in complex environmental samples [110]. Recently, SEM combined with Raman spectroscopy (SEM-Raman) provides an alternative solution by concomitantly visualising nanoplastic particles and measuring their Raman spectra, allowing identification and material analysis of nanoplastic particles. Using this method, an in vitro study successfully identified standardised 200 nm polystyrene beads premixed in dissolved sea-salt solutions and amniotic fluid [111]. Unfortunately, this method is limited by its relatively higher cost, and long duration of analysis [112].

##### Atomic Force Microscopy

Atomic force microscopes (AFM) operate on the principle of surface sensing using a sharp tip of a rigid conductive material fixed to a cantilever [113]. A recent study has demonstrated the utility of AFM in identifying submicron polystyrene particles in cultured human cells [114]. While the AFM provides better resolution than the SEM, the oscillating tip can damage the sample during measurement, leading to fragmentation and inaccurate images [112].

##### Fourier-Transform Infrared Spectroscopy

Fourier-transform infrared (FTIR) spectroscopy involves the detection of unique spectral signals released from molecules after excitation with infrared irradiation. Comparison of recorded signals against a spectral library of known plastic materials provides information regarding its composition. The theoretical limit of FTIR spectroscopy is 10 μm due to the diffraction limit of light; however, there is approximately a 35% underestimation of the number of particles even at sizes of 20 μm [115]. An additional limitation of this approach includes underestimation of particle quantities when the particles are dark coloured as the particles more readily absorb infrared signals [116,117].

##### Laser Direct Infrared Imaging System

As an alternative to FTIR, laser direct infrared (LDIR) imaging analyses microplastic particles faster than conventional FTIR spectroscopic methods. It is an infrared (IR) spectrometer utilising a fast-tunable quantum-cascade laser (QCL) as a light source which is coupled to a rapidly scanning imaging system. The instrument was originally designed for the pharmaceutical analysis of tablets, laminates, tissues, and fibres, but can also be used to analyse microplastics. Similar to an FTIR, the imaging system provides information on particle enumeration, size, and morphology, while the polymer type can be identified by the spectrometer [118].

##### Raman Spectroscopy

Raman spectroscopy is another vibrational spectroscopy technique which analyses the Raman shift of microplastic particles after irradiation with a monochromatic light source [119]. Similar to FTIR spectroscopy, Raman spectroscopy permits spectral analysis of microplastic particles in the sample and comparison against a spectral library of known plastics to identify particles. A major advantage of Raman spectroscopy is its ability to analyse particle sizes as small as 1 μm [120]. Therefore, it is used in complement with FTIR spectroscopy to identify samples smaller than 50 μm in size.

##### Limitations of Spectral Libraries

FTIR and Raman spectroscopy utilise established custom and open-source spectral libraries such as the Spectral Libraries of Plastic Particles (SLoPP) and μATR-FTIR Spectral Libraries of Plastic Particles (FLOPP) to identify microplastics [121,122]. These databases consist of spectral data of plastics analysed under pristine and standardised conditions to ensure reproducibility and accuracy of data. Unfortunately, plastic samples are often subjected to environmental factors such as heat, physical stress, chemical additives, and chemical contamination which may alter the structure and chemical composition [20]. Therefore, the spectra of environmental microplastics may be more diverse than those documented in existing spectral libraries. As an example, a study which analysed commonly used household plastic products under realistic environments with mass spectrometry obtained over 35,000 unique chemical features that were present in the material, but only 2979 compounds (8%) were identifiable based on current plastic chemical databases [88]. To overcome these issues, several groups have contributed to the construction of open-source, user-contributed spectral libraries to improve accessibility and expand the existing library of microplastic spectral signals [123,124,125]. Nonetheless, more extensive studies are required to further characterise spectral signals of microplastics to improve the accuracy and robustness of current spectroscopic methods.

#### 5.3.2. Destructive Methods

##### Thermal Analysis

In contrast to FTIR and Raman spectroscopy, pyrolysis or thermal desorption gas chromatography-mass spectrometry (Pyr-GC-MS) is an analytical method which involves the thermal degradation of large particles into fingerprint chromatograms, known as pyrograms, to assess their chemical composition. Products of pyrolysis are separated using gas chromatography and analysed with mass spectrometry to identify synthetic polymers which make up the original microplastic particles [126]. The advantage of Pyr-GC-MS over FTIR and Raman spectroscopy is its ability to quantify the masses of small microplastics (<10 μg) even with a small sample volume. However, samples analysed are destroyed as the particles are pyrolysed. In addition, although Pyr-GC-MS provides information regarding particle mass, it is unable to characterise particle quantity or structural shape or colour due to the nature of pyrolysis. Thirdly, the presence of naturally occurring polymers and additives can contribute to an overestimation of microplastic content given the similarity in structural composition [127]. Recent advances include the development of other thermogravimetric analysis (TGA)-based methods, such as TGA-FTIR, TGA-MS, and thermal extraction desorption-GC-MS [128].

Although a variety of preparatory and analytical methods are available to characterise microplastics, there is still no consensus on a standardised protocol, partly due to the limitations of current methodologies. Differences in sample preparation and analytical methodologies may limit reproducibility and comparison between studies.

## 6. Regulation of Microplastics

Legislation has been driven by environmental and human health concerns. Several countries, including the United States and European Union, have legislated to combat growing microplastic and nanoplastic contamination in the environment [129,130,131]. In 2019, the European Chemicals Agency (ECHA) drafted a proposal as part of the Registration, Evaluation, Authorisation, and Restriction of Chemicals (REACH) regulations to restrict intentionally added microplastics [131]. The United States Microbead-Free Waters Act of 2015 prohibits the manufacturing, packaging, and distribution of rinse-off cosmetics and non-prescription drugs containing plastic microbeads [130]. The state of California is the first government in the world to mandate testing of potable water for microplastics. Although legislation serves to reduce microplastic pollution by reducing their usage and production, challenges in enforceability and overly stringent regulations may jeopardise their original intentions.

Enforceability of regulations is restricted by limitations in current analytical techniques that may be unable to accurately detect and fully characterise all polymeric particles, which may be present in minute amounts. In fact, conventionally used analytical techniques may be unable to accurately identify the lower limits of particles (1 nm) outlined in the ECHA definition [112].

The lack of consensus on the definition of microplastics further confounds the usefulness of regulations. The United States Microbead-Free Waters Act of 2015 has been criticised for the limited scope of the included definition because it does not cover microbeads added to certain types of cosmetics (“leave-ons’’) and secondary microplastics produced by degradation of larger plastics, which make up the majority of microplastics found in the environment. Conversely, the original 2019 ECHA definition of microplastics was criticised as overly stringent and included all forms of polymers, including polymers functionally important to the manufacturing industry. Polymers such as derivatised celluloses, which are frequently used in the pharmaceutical sector for medication production, have also been included. Increasingly specialised drugs, such as those targeted at penetrating the blood–brain barrier, often capitalise upon the unique properties of these polymers. In many cases, there are no suitable alternatives that possess properties of the original polymer [132]. Limiting the use of these compounds may therefore hinder innovation and development of more effective therapeutics [133]. Hence, the original 2019 ECHA definition was criticised for the administrative burden it enacted upon the pharmaceutical industry while remaining ineffective in reducing microplastic pollution. Further revisions of proposed regulations in 2022 have since exempted the use of polymers in medicinal products for veterinary and human consumption [134].

It is important to note that the largest source of microplastics still arises from secondary degradation of existing plastic products and waste after natural environmental exposure [134]. Large-scale changes are therefore necessary to effectively mitigate plastic pollution and human exposure. Development of circular business models will encourage green chemistry and engineering solutions in the plastic lifecycle.

## 7. Conclusions

Plastics are abundant materials that are relatively inexpensive to manufacture. Numerous industries, including healthcare, remain dependent on plastics across a wide range of applications. Eye care, with its widespread use of topical ophthalmic formulations, surgical equipment, contact lenses, and syringes for intraocular delivery of therapeutics is no different [135]. There is mounting evidence that microplastics and nanoplastics may impact human health adversely by alterations of ocular surface immunology or microbiome, or inductions of oxidative stress or cell death. The effects of microplastics and nanoplastics on the ocular surface have yet to be determined. Research and further collaborative work in this field is paramount to ensure practitioners and stakeholders abide by the principal tenet of healthcare, ‘Primum non nocere’.

## Figures and Tables

**Figure 1 ijms-24-03928-f001:**
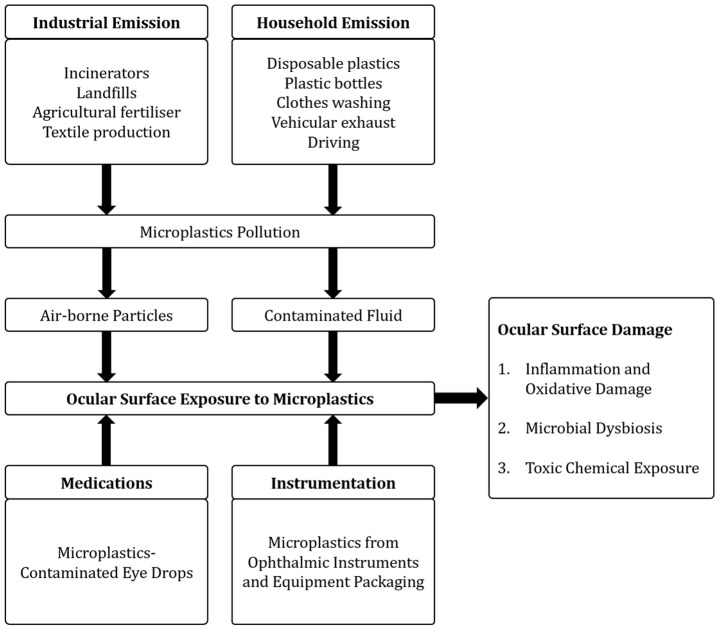
Sources of microplastics exposure.

**Table 1 ijms-24-03928-t001:** Common analytical methods for microplastics.

Analytical Method	Principle	Size Limit	Advantages	Limitations
Light Microscopy	Identification using visual, light, or digital microscopy +/− staining with Nile Red reagent	>500 μm	Quick preliminary quantification method	Subjective, assessor-dependent High probability of misidentification with smaller particles Should be complemented with other approaches
Stereomicroscopy	Microscopic views at two different angles allowing stereoscopic vision	>500 μm	Quick preliminary quantification method Allows closer visualisation of surface structure than the standard light and fluorescent microscopy	Subjective, assessor-dependent 20–70% identification rate Best complemented with other techniques
Fluorescence Microscopy	Staining with Nile Red reagent highlights microplastic particles allowing enhanced visualisation	>500 μm	Enhances visualisation of microplastics	Easily confounded by natural organic materials—thorough pretreatment is required during sample preparation
Transmission Electron Microscopy	Measurement of electrons transmitted through a sample	<1 nm	Commonly used to analyse nanoparticles	Ineffective at characterising microplastics and nanoplastics due to their electron-lucent nature
Scanning Electron Microscopy	Measurement of electrons scatter from the surface of the samples, allowing characterisation of the surface morphology and topography of the compound	<1 nm	High resolution allowing visualisation of nanoparticles Allows analysis of nanoparticles in complex environmental samples when coupled with Raman spectroscopy	Relatively expensive Long duration of analysis
Atomic Force Microscopy	Measurements of forces created between a conductive tip and the sample Has three modes: (i) contact, (ii) non-contact, and (iii) tapping	<1 nm	Provides the best resolution of particles out of all analytical tools Provides three-dimensional images of the surface structure of polymers	Inaccuracies from image acquisition may arise from fragmentation caused by mechanical stress on sample surfaces
Fourier-Transform Infrared Spectroscopy	Excitation and detection of molecular vibrational signatures via infrared irradiation	>10–20 μm	Short measurement duration Provides quantitative and qualitative information regarding each microplastic particles in the sample Non-destructive method	Very small microplastics <10 μm cannot be measured due to diffraction limit of light Significant underestimation of particles < 20 μm Thicker >100μm and blacker particles absorb infrared more strongly, resulting in underestimation of microplastic samples Cannot measure mass Identification of particles limited by existing spectral libraries
Laser Direct Infrared Spectroscopy	Infrared (IR) spectrometer utilising a fast-tunable quantum-cascade laser (QCL) as a light source	60 μm	Rapid measurement, less time consuming than Fourier-transform infrared spectroscopy	Aggregation of particles in samples may cause inaccurate readings High concentration of carbon particles may attenuate the infrared light
Raman Spectroscopy	Measurement of frequency difference in inelastically scattered photons and Rayleigh photons (Raman shift) after excitation with a monochromatic laser source	>1 μm	High reproducibility, requires low amounts of sample with minimal preparation Complements FTIR spectroscopy Non-destructive method	Duration of measurement takes >24 h Easily affected by contaminants, especially nearing size of 1 μm Identification of particles limited by existing spectral libraries
Thermal Analysis	Thermal degradation of large molecules into smaller particles to analyse their chemical composition	<10 μg	Allows for qualitative and quantification of small microplastics size Allows additional characterisation of additives	Destruction of original sample Analysis can be confounded by naturally occurring polymers (cellulose, keratin, etc.) which results in production of similar pyrolysis products, leading to overestimation of microplastics content Mass-based quantification of a non-uniform sample with a large variety of microplastics becomes complex as they are pyrolysed into similar units

## Data Availability

Not applicable.

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
