# Peer review of "Impact of Microplastics on the Ocular Surface"

_ijms, 2023, doi:10.3390/ijms24043928_

Round 1

Reviewer 1 Report

Wu and colleagues summarize current knowledge regarding the impact of microplastics on human health. The authors describe research using human, animal and cell culture systems used to investigate the effects of microplastics on cell survival, inflammation and microbiological dysbiosis. A common theme in this manuscript are the effects of microplastics on ocular tissues. In addition, the authors discuss the shortcomings of current methodology used to analyze and characterize microplastics.

Overall, this is a well written article that should be of general interest for biomedical researchers interested in environmental health issues. A few minor issues were identified:

-         Lines 88ff: This refers to the authors data suggesting microplastics in eye drops. This is a significant statement. The authors should provide data to support this statement.

-         Line 121ff: What exactly does it mean that “microplastic causes…..corneal epithelial microvilli and corneal desmosomes”?

-         Important questions regarding the presence of microplastic in cells are not discussed. Do they enter cells via endocytosis? In what form are these microplastics present in the perinuclear space? Is anything known regarding these issues? If so, this should be discussed.

-         Line 309: KOH is a base, not an acid

Author Response

Dear Editor,

We thank the reviewers for their review of our manuscript and comments. Please find our responses to the comments below:

Reviewer 1:

Wu and colleagues summarize current knowledge regarding the impact of microplastics on human health. The authors describe research using human, animal and cell culture systems used to investigate the effects of microplastics on cell survival, inflammation and microbiological dysbiosis. A common theme in this manuscript are the effects of microplastics on ocular tissues. In addition, the authors discuss the shortcomings of current methodology used to analyze and characterize microplastics.

Overall, this is a well written article that should be of general interest for biomedical researchers interested in environmental health issues. A few minor issues were identified:

We thank the reviewer for the very kind comments and agree that this is a topic of growing interest and increasing importance.  

-         Lines 88: This refers to the authors data suggesting microplastics in eye drops. This is a significant statement. The authors should provide data to support this statement.

We agree with the reviewer that this is indeed a significant statement. While our current work has demonstrated the presence of microplastics in topical ophthalmic formulations, this has yet to be published. This statement has been modified in-line with IJMS’ guidelines.

-         Line 121: What exactly does it mean that “microplastic causes…..corneal epithelial microvilli and corneal desmosomes”?

We thank the reviewer for highlighting this. The above statement has been rephrased: “which can contain microplastics, causes reduced tear volume, tear film break-up time, and destruction of corneal epithelial microvilli and corneal desmosomes”.

-         Important questions regarding the presence of microplastic in cells are not discussed. Do they enter cells via endocytosis? In what form are these microplastics present in the perinuclear space? Is anything known regarding these issues? If so, this should be discussed.

The mechanism underlying microplastic uptake is not well characterised. Intestinal uptake of microplastics is thought to occur via microvilli endocytosis and cilia movement allowing uptake in digestive tubules. While macrophages play a role in phagocytosis of microplastics, there is evidence to suggest that microplastics are incompletely broken down which may cause further metabolic dysfunction. Unfortunately, the mechanisms are not well elucidated and the existing scientific knowledge regarding the outcome of microplastics in the human body remains limited.

-         Line 309: KOH is a base, not an acid

We thank the reviewer for highlighting this. Rather than referring to KOH as an acid, the term “Acid-base solutions” was used to encompass the range of acids and bases used to dissolve organic compounds. Hydrogen peroxide, for instance, is a weak acid.

We trust that you will find these amendments satisfactory.

Best regards,

Duoduo Wu

Blanche X.H. Lim

Ivan Seah

Shay Xie

Julia E. Jaeger

Robert K. Symons

Amy L. Heffernan

Emily E.M. Curren

Sandric C.Y. Leong

Andri K. Riau

Dawn K.A. Lim

Fiona Stapleton

Mohammad Javed Ali

Swati Singh

Louis Tong

Jodhbir S. Mehta

Xinyi Su

Chris H.L. Lim

Reviewer 2 Report

This review provides a good overview of microplastics and their potential effects on the eye with further descriptions regarding methods to characterize microplastics in lab protocols. Suggestions are made to improve the quality of the paper.

-Please briefly mention the chemical identity of most microplastics currently used, as well as the reported half-life in the environment.

-The authors focus heavily on the potential pro-inflammatory properties of microplastics. Can further clarification be added regarding innate versus adaptive responses to microplastics. The assumption is that microplastics rarely induce an autoimmune response or allergic response and thus are thought to be generally innocuous to the general population.

-(lines 97-119) - It seems like all of these descriptions are referring to a single study (ref. 9). Please group into 1 paragraph and summarize the findings in a more succinct manner and reference other studies that may support these findings.

-(line 127): Please consider referencing additional human studies that have associated higher environmental microplastic exposure with ocular conditions.

-(line 225): The associations between contact lens use and disruption of the microbial microenvironment are likely more a result of hypoxia than exposure to the plastic. Please consider clarifying this within the text.

Author Response

Dear Editor,

We thank the reviewers for their review of our manuscript and comments. Please find our responses to the comments below:

Reviewer 2:

This review provides a good overview of microplastics and their potential effects on the eye with further descriptions regarding methods to characterize microplastics in lab protocols. Suggestions are made to improve the quality of the paper.

We thank the reviewer for the very kind comments and suggestions.

-Please briefly mention the chemical identity of most microplastics currently used, as well as the reported half-life in the environment.

We thank the reviewer this the suggestion. Details regarding the commonly identified microplastic in the atmospheric environment and the range of half-lives of various components have been added.

-The authors focus heavily on the potential pro-inflammatory properties of microplastics. Can further clarification be added regarding innate versus adaptive responses to microplastics. The assumption is that microplastics rarely induce an autoimmune response or allergic response and thus are thought to be generally innocuous to the general population.

Microplastics can induce both innate and adaptive immune responses. In-vitro and in-vivo studies have shown that microplastics are not innocuous to tissues – as evidenced by release of pro-inflammatory cytokines as well as evidence of histological damage of cellular tissues. These details are highlighted in section 4.1. Indeed, while plastic use is ubiquitous in daily life, our knowledge regarding its biomedical impact remains limited, and hence warrants further studies to advance our understanding in this field.

-(lines 97-119) - It seems like all of these descriptions are referring to a single study (ref. 9). Please group into 1 paragraph and summarize the findings in a more succinct manner and reference other studies that may support these findings.

This study remains, to the best of our knowledge, the only study which examined the impact of microplastics on the ocular surface in an in-vivo animal model. As this manuscript primarily explores the impact of microplastics on the ocular surface, we have placed emphasis on this study, which provides direct evidence of ocular surface damage arising from exposure to microplastics. Further studies have been cited in Section 3 detailing in-vitro and clinical studies on how particulate matter 2.5 (PM2.5), a similar environmental particulate, can cause histological, inflammatory and clinical evidence of dry eye disease and ocular surface damage.

-(line 127): Please consider referencing additional human studies that have associated higher environmental microplastic exposure with ocular conditions.

We thank the reviewer for this suggestion and have included additional studies reporting the associated effects of environmental pollution on the ocular surface.

-(line 225): The associations between contact lens use and disruption of the microbial microenvironment are likely more a result of hypoxia than exposure to the plastic. Please consider clarifying this within the text.

We thank the reviewer for this suggestion and additional details have been added in the text to reflect the associations between pressure on the ocular surface and hypoxia as suggested by Zhang et al.

We trust that you will find these amendments satisfactory.

Best regards,

Duoduo Wu

Blanche X.H. Lim

Ivan Seah

Shay Xie

Julia E. Jaeger

Robert K. Symons

Amy L. Heffernan

Emily E.M. Curren

Sandric C.Y. Leong

Andri K. Riau

Dawn K.A. Lim

Fiona Stapleton

Mohammad Javed Ali

Swati Singh

Louis Tong

Jodhbir S. Mehta

Xinyi Su

Chris H.L. Lim